# Distributed IMU Sensors for In-Field Dynamic Measurements on an Alpine Ski

**DOI:** 10.3390/s24061805

**Published:** 2024-03-11

**Authors:** Leopold G. Beuken, Joshua L. Priest, Travis Hainsworth, J. Sean Humbert

**Affiliations:** 1Paul M. Rady Department of Mechanical Engineering, University of Colorado Boulder, Boulder, CO 80309, USA; leopold.beuken@colorado.edu (L.G.B.); joshua.l.priest@colorado.edu (J.L.P.); 2Western Computer Science Program, University of Colorado Boulder, Gunnison, CO 81231, USA; travis.hainsworth@colorado.edu

**Keywords:** inertial measurement units, ski design and evaluation, outdoor testing, distributed sensors, Kalman filter

## Abstract

Modern ski design is an inherently time-consuming process that involves an iterative feedback loop comprised of design, manufacturing and in-field qualitative evaluations. Additionally consumers can only rely on qualitative evaluation for selecting the ideal ski, and due to the variation in skier styles and ability levels, consumers can find it to be an inconsistent and expensive experience. We propose supplementing the design and evaluation process with data from in-field prototype testing, using a modular sensor array that can be ported to nearly any ski. This paper discusses a new distributed Inertial Measurement Unit (IMU) suite, including details regarding the design and operation, sensor validation experiments, and outdoor in-field testing results. Data are collected from a set of spatially distributed IMUs located on the upper surface of the ski. We demonstrate that this system and associated post-processing algorithms provide accurate data at a high rate (>700 Hz), enabling the measurement of both structural and rigid ski characteristics, and are robust to repetitive testing in outdoor winter conditions.

## 1. Introduction

Alpine skiing is a highly popular winter sport that not only generates significant financial turnover for ski resorts and equipment manufacturers (2023 generated USD 4.6bn in the US alone [1]), but also creates considerable consumer demand for optimally performing, safe, and reliable equipment [2]. Despite this large market demand for performance equipment, manufacturers often rely on small, incremental improvements [3,4] in ski design through trial-and-error testing and development methods that typically have slow design cycles [2,3,4]. Furthermore, it is often very challenging for consumers to efficiently and accurately select ski equipment based on their personal style and snow conditions [5,6].

It is common for both manufacturers and end-users to rely on qualitative data to assess the equipment in question. For manufacturers, this may result in slow or inefficient design cycles, expensive product development phases and incremental improvements in ski technology. For consumers the use of qualitative data makes the ski selection process difficult, with many factors introducing uncertainty in the equipment selection process. Furthermore, qualitative data from expert equipment reviewers (that consumers often rely on to make decisions) may not be appropriate for a consumer with different skills, attributes or needs. Quantitative data for these decisions are typically not available due to the challenge of utilizing sensors in a harsh environment such as alpine skiing. We propose addressing this quantitative gap through a novel sensor suite agnostic to skier or ski type. This paper discusses the design and implementation of a dynamic sensing tool that is located directly on the upper surface of an alpine ski, consisting of a set of 10 distributed IMUs collecting data at a high rate. Additionally, this system has proven to be robust to the harsh conditions found during alpine skiing.

In related studies aimed at improving the efficiency of the ski design process, researchers have explored various domains to develop insight into the in-field characteristics and behavior of the ski. The use of high-fidelity numerical simulations has been investigated as a method that can complement traditional ski design cycles [2,3,4,5,7,8]. In this domain, researchers can simulate a range of equipment designs to determine favorable characteristics before the manufacturing and testing steps take place in the design cycle. A concern with simulations is that the accuracy relies on accurate ski loading information, which is challenging to determine without in-field measurements. Besides simulations, laboratory testing setups have also been devised to measure static stiffness profiles of skis [9,10] or binding deformation shapes [11], which can also be used as an indicator of ski performance before qualitative in-field testing takes place [9,11,12]. Dynamic laboratory testing has been performed on cross-country skis using a custom-made tribometer [13,14] and “skitester” [15]. These experiments serve as a proxy for in-field testing, as equipment can be used in a controlled, repeatable environment.

Aside from a direct analysis of the skis, quantitative data have been collected from in-field experiments via sensors located on various locations on the skier’s body. IMU suits are a popular sensing modality in studies that focus on vibration [16] or turn detection [17]. Other studies investigated skier turn detection algorithms for alpine skiing and utilized a small number of IMUs across various locations on the skier body such as the knee [18] or boot cuff [19,20]. Ref. [6] had a similar sensing setup and used in-field data to classify specific skiing maneuver types. Ref. [21] used the IMU found inside a smartphone to track a skier through uncontrolled, in-field testing. The in-field sensing systems mentioned above provided valuable insight into skier performance, but were not able to investigate the dynamic motion of the ski equipment.

Fewer studies have utilized sensors positioned directly on the ski equipment and not on the skier’s body. Ref. [22] used both inertial sensors and a 6-axis force sensor on the ski binding for skier turn assessment. Refs. [23,24] used IMUs on ski poles to determine pole angles and forces during downhill, in-field ski runs. Plantar boot pressure sensors were utilized by [25] to analyze the carving technique. These studies gave insight into the skier but not necessarily the ski equipment.

With regards to studying the ski: studies have investigated the strain distribution on an alpine ski for in-field testing, but without concern for inertial measurements. Refs. [26,27] used resistive strain gauges, while [28] developed a custom-made sensor to measure continuous ski curvatures during downhill ski runs. Ref. [29] utilized a similar resistive strain gauge sensor setup to classify snow conditions and, finally, Ref. [30] fixed pressure pads to the edge of the ski to determine the forces acting on the ski during carving turns. All these studies utilized some form of sensor located directly on the ski equipment itself for either skier evaluation or equipment testing.

The works mentioned in this paragraph are most similar to what is presented in this paper but have fewer inertial sensors, lower sampling rates and/or have not been shown to be robust for in-field testing. Ref. [31] implemented a set of five independent accelerometer modules at spatially distributed locations on the ski to measure both bending and torsional effects. Ref. [32] had a similar sensing system that benefited from wireless technology for data transmission to a central microcontroller. Ref. [33] used two or more accelerometers attached to the ski shovel to measure both the bending and twisting motion during in-field runs. A set of nine distributed IMUs were placed on the upper surface of an alpine ski by [34], who performed limited in-field testing and mainly focused on laboratory vibration testing on an ice bank. The three goals of our research are to develop a sensing tool that has distributed inertial sensing of the structural dynamics of an alpine ski; can collect data at a high rate; and is robust to the harsh conditions experienced during in-field testing. All of the works listed can achieve one or two of these objectives, but none can achieve all three simultaneously.

The major contribution of this work is the development of a custom IMU sensor suite that is located directly on the upper surface of an alpine ski. This system is characterized by 10 spatially distributed IMU sensors that are polled at a high rate (>700 Hz) in order to accurately record the structural dynamics across the entire length of the ski. The sensor suite is specifically designed for extensive outdoor, in-field testing and has been shown to be highly robust in terms of both electrical and mechanical characteristics. Furthermore, the sensing tool discussed in this paper utilizes state-of-the-art micro-electromechanical (MEMS) inertial sensors, is portable, simple to manufacture, and is relatively low-cost. This paper details the design and implementation of the sensor suite along with results of both laboratory validation and in-field data collection and processing. The end goal is to provide a robust system and method to measure the dynamical structural characteristics of a ski with the potential to provide both manufacturers and consumers with relevant quantitative data to make better informed decisions in regard to design or purchasing.

The remainder of the paper is organized as follows: Section 2 addresses the design and implementation of the distributed inertial sensing system and the testing methodology for this system is detailed in Section 3, both for laboratory validation and in-field testing. Section 4 covers the data processing algorithms used and Section 5 presents detailed results from both laboratory validation testing and in-field data collection. Finally, the results are discussed and potential future experiments are introduced in Section 6, followed by a brief conclusion in Section 7.

## 2. Materials and Methods

This section details the technical design and implementation of the distributed sensor array on the alpine ski (Bonafide 2018/2019 in 180 cm, Blizzard-Technica, Treviso, Italy). The sensor suite’s design requirements are discussed in Section 2.1, followed by two Section 2.2 and Section 2.3 that further describe the electronic and mechanical components, respectively. Although this system is attached to a specific ski for experimentation purposes, it can easily be ported to other models for similar testing, as no components are permanently fixed to the ski. The sensing system consists of ten IMU sensors spatially distributed along the length of the ski and a Central Processing Hub (CPH) (Figure 1A,B) positioned in front of the ski binding.

### 2.1. Design Requirements

This section details some of the design requirements for the system, with the most important requirement being that the sensing system is highly robust to the harsh conditions experienced during in-field testing. This means that the system needs to be water-resistant and also will require mechanical protection due to external impacts that are likely to occur. Additionally, the system needs to be firmly attached to the ski but still needs to be portable to other equipment. It is critical that the system is lightweight, as to not affect natural ski/skier behavior and dynamics during in-field testing. Due to the rich spatio-temporal nature of the forces and dynamics the ski is expected to experience outdoors, a minimum of 10 spatial locations need to be sampled to generate an accurate depiction of the structural dynamics experienced.

Ref. [34] suggests that the IMUs may experience upwards of 150 G in terms of acceleration, which means that conventional MEMS devices may not be feasible. The use of more advanced inertial sensors with greater measurement ranges would make this project prohibitively expensive. Contrarily, Ref. [31] reported acceleration values well below what [34] hypothesize to be maximum acceleration values, indicating that conventional MEMS accelerometers may indeed be suitable to measure the structural dynamics of a ski during in-field testing. Typical measurement ranges of MEMS gyroscopes are within the range of angular rates expected from the ski’s structural dynamics and our results never saturate the sensors, supporting [34]’s hypothesis.

The sensing rate required of a system is typically determined by the phenomena the sensors are expected to measure. According to the Nyquist Criterion, one needs to sample a phenomena at a minimum of twice the highest frequency of interest; however, it is generally accepted that a 5–10 times faster sampling rate allows a more accurate reconstructions of the true signals being measured and the Nyquist rate is considered a theoretical minimum. The first bending mode of alpine skis is close to 20 Hz, but the second mode may occur at frequencies up to 100 Hz [33,34], meaning that a minimum sampling rate of 200 Hz is required (according to the Nyquist Criterion), but a sampling rate of around 500–1000 Hz is ideal to accurately measure the dynamical phenomena of the ski. Furthermore, the use of a higher sampling rate also opens the possibility to measure higher order structural modes that existing methods have not been able to capture due to inadequate sampling rates. Our system falls within this range with a maximum sampling rate of approximately 730 Hz.

It has to be noted that the use of inertial sensors to measure the structural dynamics of an alpine ski during outdoor testing is a relatively unexplored research domain; hence, it is challenging to determine exact specifications for sensors as the structural phenomena are not fully understood. Our work aims to develop a sensing tool that expands the operating envelope of inertial sensing for alpine skis both in terms of sensing rate and number of sensors, allowing the measurement of dynamical phenomena in a way that was not previously possible.

### 2.2. Electronics

The electronic implementation of the distributed sensor suite consists of a combination of off-the-shelf and custom-designed components. The ten IMUs (MPU-42688-P, TDK InvenSense, San Jose, CA, USA) are located directly on the upper surface of the ski along the *x*-axis (Figure 1C). These are high-quality MEMS sensors that are typically used in avionics packages, robotics and virtual reality. Some of the specifications are listed in Table 1. The data from the IMUs are aggregated in the CPH which is positioned in front of the ski binding (Figure 1A,B) and is home to all components other than the IMU sensors.

The central processing hub is home to the system power and data acquisition electronics. This system consists of a microcontroller (MCU) (Teensy 4.1, PJRC, Sherwood, OR, USA) that is responsible for data aggregation and storage and also serves as the central controller for all other components (Figure 2A). All sensors are controlled by the MCU, which results in a synchronized clock source and removes the need for synchronizing multiple sensors from various MCUs post data collection. The CPH also houses the signal and power distribution board that is custom designed and serves various functions: the board regulates the battery voltage (7.2 V) to a usable 5 V that powers the MCU as well as all IMU breakout boards and arranges relevant power lines and communication signals into an appropriate format to be distributed by a ribbon cable (see Figure 2B). Additionally, the board contains components that are critical to signal integrity such as decoupling capacitors on power lines and serial terminating resistors on signal traces. User control switches are interfaced with this board to control basic functionality such as power and data acquisition toggling. Finally, the CPH houses the battery (7.2 V, 2.25 AH, Jauch Quartz America Inc., Seattle, WA, USA) that powers all components and has been shown to operate upwards of twelve hours during sub-freezing temperature field testing. The lithium-ion battery is specifically selected because it can deliver sufficient power at the low temperatures that are seen during in-field testing.

The MCU is responsible for data collection from all sensors via two separate serial peripheral interface (SPI) buses, each collecting data from half of the IMU sensors. The theoretical maximum data collection rate across all sensors is 138 Hz using the I2C communication protocol, but is much higher using the Serial Peripheral Interface (SPI) protocol because addressing is not required on data lines and higher clock speeds are permitted. High SPI clock speeds result in unreliable data transmission on this system and as a result, the SPI buses operate at the lowest possible clock speed of 1 MHz; however this does not limit the overall sensor suite’s data rate which is in the kHz range. Furthermore, the MCU generates timestamps associated with each sensor reading and writes data to a micro secure digital (SD) card through the built-in reader/writer. Testing revealed that the MCU clock yielded accurate timestamp information (<0.1% drift), negating the need for a real-time clock for timekeeping. Sensor sampling rate is set through the custom firmware running on the MCU and exceeds 700 Hz while simultaneously writing data to the SD card. These rates can be increased further if data are not written onto the SD card, but instead streamed to a desktop computer, as may be the case when performing benchtop experiments. Finally, all data transmission occurs physically over a 14 channel ribbon cable (Figure 2B) that transfers power, SPI signal lines and all chip select lines for the SPI bus. Signal cables are separated with ground lines to reduce electromagnetic interference.

The IMU sensors are located on custom breakout printed circuit boards (PCBs) (Figure 2C) that have a small form-factor and simple interface with the ribbon cable carrying power and SPI signal lines. Each breakout PCB contains a linear drop-down voltage regulator that generates 3.3 V to power the IMU and every breakout board is connected to all chip select (CS) lines associated with the particular SPI bus being used. A jumper resistor from one of the CS lines is used to select a particular IMU breakout board in what is effectively a multiplexing scheme. All SPI lines (clock, peripheral-out-controller-in (POCI), peripheral-in-controller-out (PICO), chip selects) are passed through the length of each breakout board (and also branched to the IMU chip) to an opposite ribbon cable connector used to connect the subsequent breakout board in the chain. Standard 0.1 inch pins provide the user access to SPI signals to aid in system debugging.

### 2.3. Mechanical Design

The IMU sensors and accompanying electronics are subject to cold and wet conditions during in-field testing and may also experience incidental, external mechanical shock from the opposite ski, debris on the course or even ski poles. This necessitates physical protection that is both robust mechanically, and also water-resistant. To address these design constraints, all electronic components are housed in 3D-printed parts manufactured with thermoplastic polyurethane (TPU) material (PolyFlex TPU90, Polymaker, Houston, TX, USA). The covers serve a dual purpose: firstly, they protect electronics from external mechanical shock, and secondly, they provide a water resistant operating environment. The IMU sensor housings consist of two components—the base ring and the dome cover. The base ring is attached to the ski with a double-sided adhesive film (3M, 9474LE 300LSE) and the cover is attached to the ring with press-fit knobs (see Figure 1D) and sealed with a water-resistant gasket sealant (Optimum Black Gasket Maker, Permatex, Solon, OH, USA). The central processing hub has a similar protective setup that only differs in shape and size to the domes that cover the IMU sensors. The ribbon cables that transfer signals from the sensors to the central processing hub are attached to the ski with the same double-sided film as the covers and are further protected with a layer of highly adhesive tape (EXTREME HOLD Duct Tape, 3M, St. Paul, MN, USA).

A central design requirement for the sensing system is that the addition of sensors should have minimal effects on both the ski and skier. The covers were manufactured from a flexible material to limit the mechanical effect on the ski, creating minimal stress concentrations on the ski surface which not only enables the ski to behave naturally during testing, but also allows the cover to flex along with the ski. This flexibility maintains a consistent ski-housing interface instead of separating, as a more rigid cover would tend to do over time due to fatigue caused by vibrations. Furthermore, the covers are designed to have a low, curved profile to withstand glancing, external blows.

All components are lightweight to minimize inertial effects on the ski. The nominal weight of the ski is 3.33 kg and the addition of the sensing equipment raises the overall weight to 3.74 kg, a 0.41 kg increase (12.3%). This small increase can be considered negligible because the majority of the additional weight is located near the center of gravity of the ski in the CPH, limiting the rotational inertia created by the additional mass.

## 3. Testing Methodology

The sensing system is evaluated in two distinct settings. Firstly, the sensors are tested in a laboratory environment where sensor readings and Kalman filter orientation estimates are compared to ground truth data from a visual motion capture system in quasi-static, dynamic and impulse tests. The distributed IMU system is also extensively tested outdoors in typical skiing conditions to determine the robustness of the sensors and typical data that can be expected from in-field testing.

### 3.1. Laboratory Sensor Validation Testing

Laboratory testing is performed in an indoor setting to validate IMU sensor readings. Sensor readings were validated through comparison to what is considered ground truth data collected from a visual motion capture system. This system consists of eight cameras (Prime 13W, OptiTrack, Corvallis, OR, USA) generating pose (position and orientation) information at 240 Hz with an accuracy of ±0.2 mm. When the “Record” switch on the ski is toggled, an infrared LED connected to the CPH is powered simultaneously. Motion capture cameras are able to detect the LED when it switches on, resulting in a simple method of accurately time syncing the IMU readings and motion capture data. Three distinct validation experiments are performed: a quasi-static test and two dynamic movement tests (a general dynamic movement test and an impulse test).

The quasi-static test is designed to evaluate the accuracy of Kalman filter (KF) orientation estimates. Dynamic, inertial IMU readings are used to estimate the static orientation of the IMU using the KF algorithm (discussed in Section 4), hence the requirement for a static orientation experiment where the ski is slowly moved to various angles between −90° and +90° along each of the three axes (roll, pitch and yaw). This test not only allows us to evaluate the performance of the KF, but also provides an indication of the quality of data received from the IMU; the KF will yield poor orientation estimates when presented with inaccurate inertial data from the IMU.

The dynamic test involves manually-generated, dynamic movements of the ski in the motion capture space and consists of two separate styles of movement: perturbation in both a rotational and linear manner along each of the three axes. The motion capture software only provides pose information (position and orientation), and additional processing is required to obtain values that are directly comparable to raw IMU data. Rotational motion capture data are numerically differentiated and rotated to the sensor frame for direct comparison to gyroscope data. Linear data are twice-differentiated and rotated in a similar manner to the rotational data, but require an additional processing step where an artificial gravity vector is introduced, after which the linear data are in the same frame as the accelerometer. Motion capture data are low-pass-filtered (5 Hz cutoff frequency) to remove any high-frequency noise that would be accentuated during the numerical differentiation schemes.

The impulse test is designed to more closely recreate vibrational phenomena that the ski may experience during in-field testing. Here, the ski is clamped to a rigid table with the front half cantilevered over the table edge. The ski tip is deflected approximately 8 cm and abruptly released, resulting in a vibratory response of the free end of the ski. This represents the natural vibration of the ski, a critical aspect that the distributed IMU measurement tool is expected to measure during in-field testing. Data are processed in a similar way as for the dynamic movement test where the vertical acceleration measurement is compared to the estimated vertical acceleration estimates from motion capture data. The movement of a rigid body object surrounding the second IMU (from the ski tip) is tracked.

### 3.2. In-Field Testing

In-field testing took place on various days at different ski resorts in Colorado, USA. The sensing system is attached to the right ski of both an intermediate (175 cm, 75 kg) and an expert skier (170 cm, 69 kg) while each performed a variety of maneuvers and skied slopes of varying difficulty. Datasets from more than 15 ski runs were collected over the course of in-field ski testing experiments. The skiers were aware of the purpose and goals of this experiment and agreed to the data collection methods used. A recording session consists of (1) powering the sensing system with the control switch (the ski is required to be stationary for 60 s for the gyroscope initialization scheme to be completed), (2) ensuring the skier is free to roam the ski slope until data are to be recorded (typically immediately before the initiation of a run) when they can toggle the recording switch to initiate data storage and (3) ensuring that when a run is completed, the recording switch is toggled to complete data collection. Multiple data collection periods can be initiated during any power cycle and the system can store data across different power cycle sessions. All data are analyzed post data collection and collected from the onboard SD card.

## 4. Data Processing—Kalman Filter

A Kalman filter (KF) implementation (Navigation Toolbox, MATLAB 2023a [35]) is used to estimate orientation from IMU data. The KF is a mature sensor fusion algorithm that provides optimal state estimates of the sensor’s orientation (in the least squares sense) for linear systems [36]. The ten IMUs each provide six pieces of dynamic information: linear acceleration and angular velocity measurements along a local, right-handed coordinate frame oriented in the east–north–up (ENU) configuration. For this algorithm, the raw data measurement coordinates are rotated into the north–east–down (NED) coordinate frame (see Figure 1C) for processing by the KF. This section provides a high-level overview of this algorithm, with additional details available at [37,38]. Note that IMU sensor data are post-processed after data collection and do not run onboard the ski MCU.

The KF implemented in this study is more precisely defined as an Indirect Complementary KF where the term *Indirect* indicates that the filter operates on an error state vector (as opposed to the absolute state vector) and *Complementary* refers to fusing orientation estimates from both the gyroscope and accelerometer components of the IMU readings. Because of these changes to the traditional KF approach, there are some additional steps required to generate absolute orientation estimates. Figure 3 shows the overall block diagram for the KF implementation, where each labeled block is discussed in Section 4.1–Section 4.3.

### 4.1. Error Measurement Generation—Block A of Figure 3

The KF implementation operates on an error state vector that is combined with previous absolute estimates to generate current absolute orientation estimates. First, a pseudo error measurement vector is generated from raw, absolute IMU measurements, as seen in Block A (Figure 3A). Bias-corrected gyroscope angular rate readings, $z_{G,k}^{\prime }=z_{G,k}-b_{k-1}^{+}$ (where $z_{G,k}$ is the current gyroscope reading and $b_{k-1}^{+}$ is the estimated gyroscope bias at the previous time step) are numerically integrated and added to the previous absolute orientation estimate, $\theta_{k-1}^{+}$, to form, $\theta_{k}^{-}$, the current a priori absolute orientation estimate. For clarity, the subscripts in this block are defined as follows: *G*—quantity derived from gyroscope reading; *k*—current reading/estimate; k−1—previous reading/estimate. Superscripts with “^+^” refer to a posteriori estimates, “^−^” superscripts represent a priori estimates, and subscripts with “*e*” characters refer to error states.

$\theta_{k}^{-}$ is used to generate an estimate of the gravity vector in the sensor frame, $g_{G,k}^{-}$. Concurrently, body acceleration estimates (in the world frame) from the previous time step, $a_{k-1}^{+}$, are subtracted from current accelerometer readings, $z_{A,k}$, (in the sensor frame), $z_{A,k}^{\prime }=z_{A,k}-a_{k-1}^{+}$, and is used to generate another estimate of the gravity vector, $g_{A,k}^{-}$. The combination of these results in $z_{e,k}=g_{G,k}^{-}-g_{A,k}^{-}$ and represents the error measurement input to the KF equations in Block B. Furthermore, $g_{G,k}^{-}$ is used in block B to calculate the dynamic error measurement matrix as seen in Equation (Equation 8).

**Figure 3 sensors-24-01805-f003:**
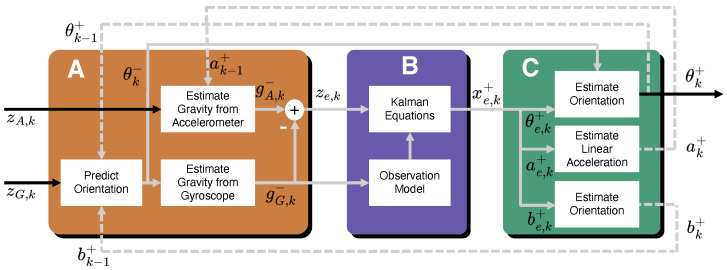
Data flow diagram of the complementary, indirect Kalman filter used for attitude estimation from IMU data. (**A**): Error measurement information is generated through gravity vector estimation from both accelerometer and gyroscope data, hence the *complementary* filter. (**B**): Indirect Extended Kalman filter equations which operate on attitude error estimations. (**C**): Absolute attitude estimation based on error signals from block B. Note that feedback signals from a previous time step are shown with a dashed line.

### 4.2. Indirect Kalman Filter—Block B of Figure 3

In this block, the indirect KF equations are implemented (which are operating on state error estimates, rather than the absolute state estimates). The KF produces posterior error state estimates defined by $x_{e,k}^{+}={\left[\theta_{e,k}^{+}\;b_{e,k}^{+}\;a_{e,k}^{+}\right]}^{T}$, where $\theta_{e,k}^{+}\in R^{3\times 1}$ is the error in orientation estimate at time *k* and expressed as Euler angles, $b_{e,k}^{+}\in R^{3\times 1}$ is the gyroscope zero angular rate bias error and $a_{e,k}^{+}\in R^{3\times 1}$ is the body acceleration error vector as seen in the world frame. The conventional KF equations are applied in this block with one small simplification: the a priori error state estimate, $x_{e,k}^{-}$ is assumed to always be zero (because the filter operates on error states), resulting in the following KF equations
(1)$${\hat{x}}_{e,k}^{-}=0$$
(2)$${\hat{P}}_{e,k}^{-}=Q_{k}$$
(3)$$y_{k}=z_{e,k}$$
(4)$$S_{k}=H_{k}P_{e,k}^{-}H_{k}^{T}+R_{k}$$
(5)$$K_{k}=P_{e,k}^{-}H_{k}^{T}{\left(S_{k}\right)}^{-1}$$
(6)$$x_{k}^{+}=K_{k}y_{k}$$
(7)$$P_{e,k}^{+}=P_{k}^{-}-K_{k}H_{k}P_{k}^{-}$$
where Table 2 details terms that have not been previously defined. $H_{k}$ is the error observation matrix relating error measurements to error states and is a function of the gravitational vector derived from gyroscope readings (in Block A)
(8)$$H_{k}=\left[\begin{array}{ccccccccc} 0 & g_{G,k,z}^{-} & -g_{G,k,y}^{-} & 0 & -\kappa g_{G,k}^{-} & \kappa g_{G,k,y}^{-} & 1 & 0 & 0\\ -g_{G,k,z}^{-} & 0 & g_{G,k,x}^{-} & \kappa g_{G,k,z}^{-} & 0 & -\kappa g_{G,k,x}^{-} & 0 & 1 & 0\\ g_{G,k,y}^{-} & -g_{G,k,x}^{-} & 0 & -\kappa g_{G,k,y}^{-} & \kappa g_{G,k,x}^{-} & 0 & 0 & 0 & 1 \end{array}\right]$$

Here, $g_{G,k,x}^{-}$, $g_{G,k,y}^{-}$ and $g_{G,k,z}^{-}$ are the components of the gravity vector estimated from gyroscope readings and κ is a parameter determined by the sample rate of the gyroscope in relation to the accelerometer and is set to one when these sample rates are equal.

### 4.3. Absolute a Posteriori State Estimates—Block C of Figure 3

Block C receives the a posteriori error state vector estimates, $x_{e,k}^{+}$, from the KF and uses them to update the absolute state estimate values. A posteriori gyroscope bias, $b_{k}^{+}$, and body linear acceleration estimates, $a_{k}^{+}$, are simply calculated as
(9)$$\left[\begin{array}{c} b_{k}^{+}\\ a_{k}^{+} \end{array}\right]=\left[\begin{array}{c} b_{k-1}^{-}\\ a_{k-1}^{-} \end{array}\right]-\left[\begin{array}{c} b_{e,k}^{+}\\ a_{k,k}^{+} \end{array}\right]$$
where current error estimates are subtracted from absolute estimates at the previous time step.

The absolute, a posteriori orientation quaternion is calculated by multiplying the a priori orientation estimate, in quaternion form, by the error in orientation output from block B (note that multiplication of quaternions result in composition of their respective rotations),
(10)$${\left(\theta_{k}^{+}\right)}_{q}={\left(\theta_{k}^{-}\right)}_{q}{\left(-\theta_{e,k}^{+}\right)}_{q}$$
where the “*q*” subscript indicates that orientations are in quaternion form.

## 5. Results

This section presents data obtained from the distributed IMU sensing system in both laboratory and in-field testing environments. Section 5.1 discusses the quasi-static and dynamic lab validation results, while Section 5.3 covers the results from in-field testing.

### 5.1. Laboratory Validation Testing

In this lab validation experiment, the ski is manually moved in a quasi-static manner in the motion capture space to test the efficacy of the KF algorithm. Figure 4 shows the results of this experiment where the KF-estimated orientation angles are shown on the same plot as the motion capture orientation estimate. The difference between the two orientation estimates defines the error also shown in the second row of plots in Figure 4. Note that only data from the single IMU under the ski boot are used for this experiment, as the purpose is to test the orientation estimation capabilities of the KF. IMU data are collected at 710 Hz and motion capture data at 240 Hz. Linear interpolation is used to generate error plots between the two sensing systems with different sampling rates, where IMU data (710 Hz) are interpolated linearly to match the timestamps from the motion capture system (240 Hz).

We see excellent tracking around each of the three axes with errors that are bounded by ±3 degrees for the majority of the test. The correlation coefficient, R, between KF estimates and motion capture pose data are 0.99 for each of the axes, indicating highly accurate KF estimates when compared with ground truth motion capture data. There are deviations from the ground truth during periods when the ski is moved to a new orientation angle—an expected result because the KF estimates take a small amount of time to converge to the new orientation. Upon closer inspection of the error in roll angle estimates around the 57 s timestamp, it is determined that IMU data are not saved to the onboard SD card for a interval of 20 ms, resulting in a short period of inaccurate estimates. This is a good illustration of the robustness of the KF algorithm that is able to maintain reasonably accurate forward estimates even when current sensor data are unavailable.

We also see small constant offsets for very large pitch angles (around ±90°); however, these orientations are highly unlikely to be experienced by the ski during in-field testing. Note that the KF performs remarkably well in estimating the yaw angle. The yaw angle is typically challenging to estimate over long time periods because it is calculated through numerical integration of the gyroscope measurements about the yaw axis (without absolute correction from the accelerometer readings), leading to potential estimation drift due to gyroscope biases. Minimal drift is observed during this test, an indication of small biases in gyroscope readings. This experiment shows not only that the KF implementation can accurately estimate the orientation of individual IMUs, but also that the IMUs installed on the ski are providing the KF algorithm with accurate sensor measurements.

The dynamic testing experiment is primarily concerned with dynamic movement of the ski and the corresponding IMU readings. The ski is manually moved in the motion capture space along each of the three axes in both a rotational and linear manner for the analysis of gyroscope and accelerometer data, respectively. Figure 5 and Figure 6 show the results for this testing, where Figure 5 shows a comparison between accelerometer data from the IMU and calculated dynamic values from the motion capture system, and Figure 6 shows the same information for gyroscope data. The correlation coefficients between the gyroscope and motion capture angular rate data are 0.994, 0.997 and 0.993 around each of the three axes and 0.997, 0.989 and 0.991 for the linear accelerometer data. This shows excellent agreement between raw IMU data and information from the motion capture system which is considered ground truth. Although IMU measurements generally follow the values calculated from motion capture data for both gyroscope and accelerometer readings, there is a small difference in amplitude at higher frequencies, especially for linear acceleration. These discrepancies are likely due to slight misalignment in axes between the ski and motion capture system. Also note that although the motion capture data are considered the ground truth in this experiment, there may be small errors associated with its estimations which may lead to some minor discrepancies in the results.

The impulse test is designed to more closely imitate ski vibrations that are present during actual in-field testing. Figure 7A shows the response (vertical z-acceleration) for all five IMUs on the forebody of the ski that are excited by a decaying sinusoidal vibrational motion. A shorter excerpt from these data are presented in Figure 7B, where the differing amplitudes of response are seen for each IMU. It is clear that IMUs located closer to the tip of the ski (refer to the legend at the top of the figure) experience higher accelerations, an expected result considering that sensors closer to the ski tip experience larger movements and, hence, larger accelerations. It is also apparent that there appears to be additional higher frequency content in the early stages of the vibratory response, which is likely the excitation of higher vibrational modes of the ski. Figure 7C shows the comparison of IMU vertical acceleration data with motion capture data and Figure 7D shows the error between these datasets (calculated with linear interpolation as for other validation tests). It is apparent that significant error exists between values reported by the IMU and ground truth motion capture values, although it appears that these errors are induced by a phase shift and magnitudes appear to be very comparable. This phase error is discussed in detail in Section 5.2.

### 5.2. Cross-Correlation Analysis of Dynamic Validation Experiments

A close inspection of the errors generated from the two dynamic experiments (manual dynamic movements and impulse test) shows generally periodic waveforms. Due to the periodic nature of the signals that these errors originate from, it is likely that some of the error is introduced due to a phase shift in data as can be seen in the results of the impulse test in Figure 7C. To examine this phase shift further, the cross correlation is calculated between IMU sensor readings and dynamic estimates from the motion capture system, providing an indication of the magnitude of phase shift present in data. Motion capture data are linearly interpolated to match the timestamp information associated with the IMU sensors to maximize the temporal resolution of this analysis. Figure 8A shows a stem plot where the correlation of the signals is compared at various integer timeshifts of the data for the impulse test dataset (this plot is typical for all dynamic validation datasets evaluated). This figure shows that maximum correlation occurs at approximately −9 or −10 integer shifts of the motion capture timescale. The detailed results of the cross-correlation analysis can be found in Table 3, where it shows that across all dynamic validation experiments, a phase shift of −9 or −10 (approximately 12.85 to 14.29 ms) timestamps is required to maximally align the signals. The table also shows the R value of the original signals and improvement in R value of signals that are shifted to maximize correlation, along with the percentage improvement. A significant reduction in error is clear for all dynamic tests when signals are phase-corrected, as can be seen in Figure 5, Figure 6 and Figure 7E. The phase error is insignificant for the quasi-static test as movements are very gradual.

**Figure 5 sensors-24-01805-f005:**
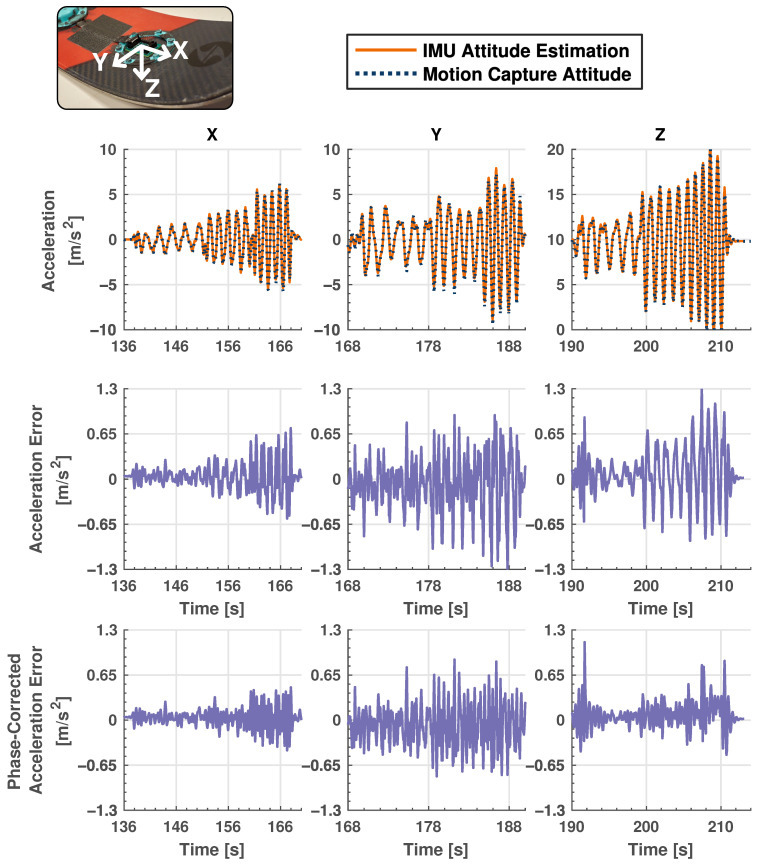
Dynamic validation testing results representing absolute and measurement error results between linear acceleration IMU and motion capture data. Solid orange plots represent IMU data, while dashed, navy blue lines show motion capture data. Two separate error plots are shown: the first shows error between IMU and motion capture data and the second shows the error when the phase of the motion capture data is corrected to account for measurement latency. Note the scaling difference between absolute and error plots. The insert is for reader reference to the relevant local body axes. IMU sampling rate: 710 Hz. Motion capture sampling rate: 240 Hz.

**Figure 6 sensors-24-01805-f006:**
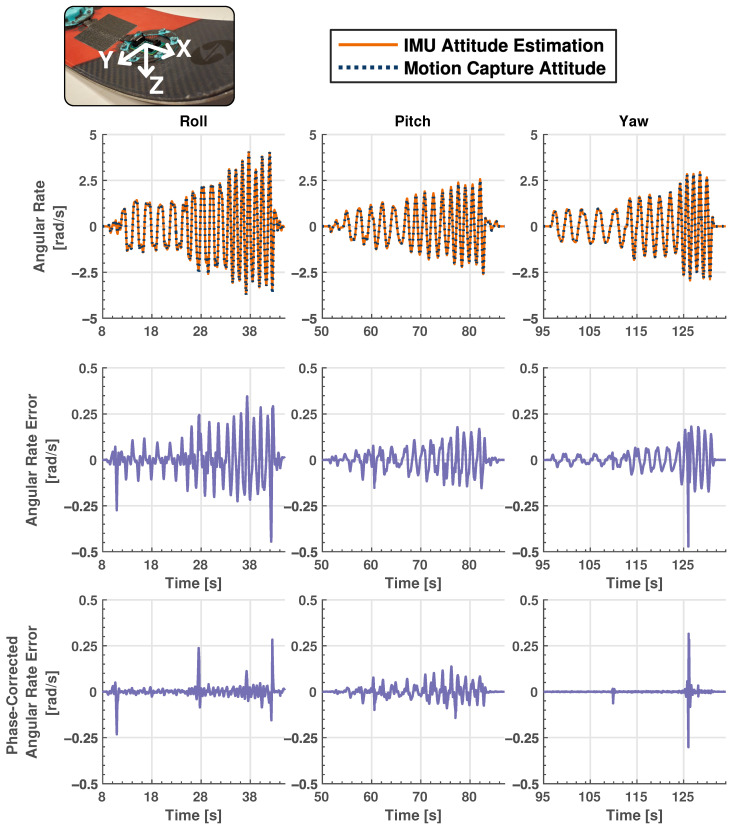
Dynamic validation testing results representing absolute and measurement error results between angular rate IMU and motion capture data. Solid orange plots represent IMU data, while dashed, navy blue lines show motion capture data. Two separate error plots are shown: the first shows error between IMU and motion capture data and the second shows the error when the phase of the motion capture data is corrected to account for measurement latency. Note the scaling difference between absolute and error plots. The insert is for reader reference to the relevant local body axes. IMU sampling rate: 710 Hz. Motion capture sampling rate: 240 Hz.

**Figure 7 sensors-24-01805-f007:**
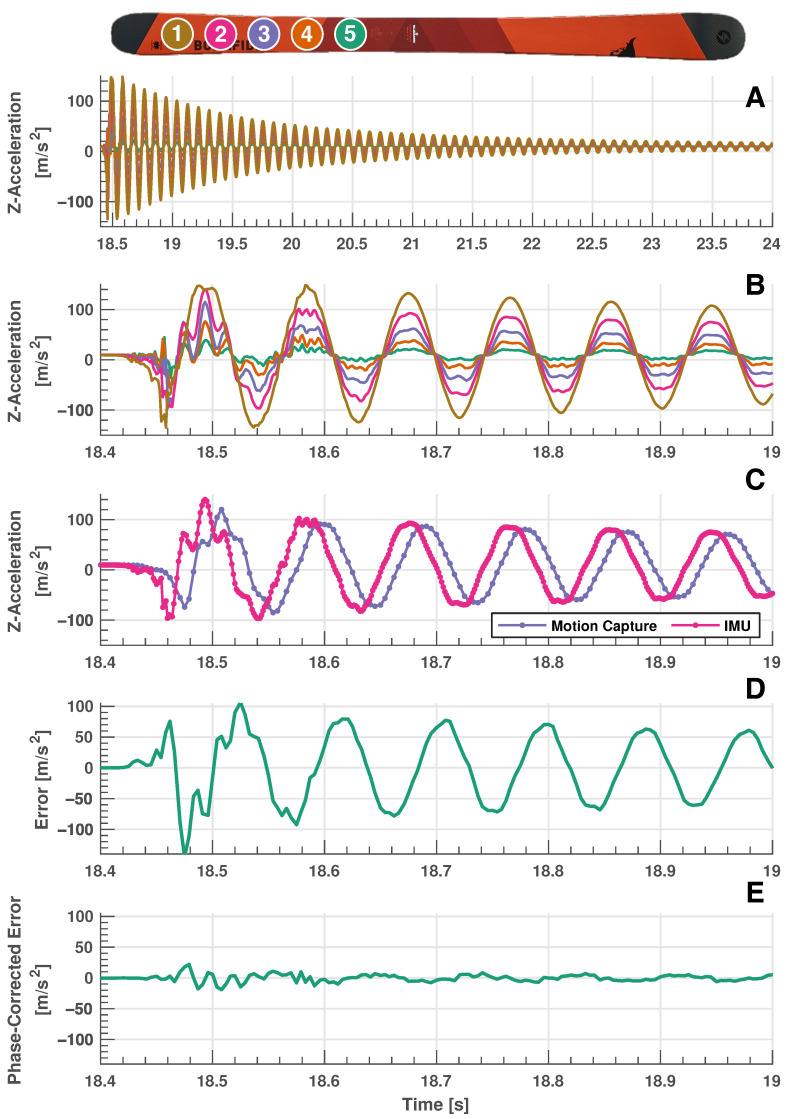
Impulse validation testing results. (**A**): Decaying sinusoidal response of the 5 IMUs that are excited by the forebody ski vibration. (**B**): Same data as subfigure (**A**), but represented on a smaller timescale. Brown line: IMU 1, magenta line: IMU 2, purple line: IMU 3, orange line: IMU 4, green line: IMU 5 (IMUs listed from tip to binding). (**C**): Comparison of IMU z-acceleration data with motion capture estimated acceleration. Magenta line: IMU data, purple line: motion capture data. (**D**): Error between IMU z-acceleration and motion capture estimated acceleration. (**E**): Error between IMU z-acceleration and motion capture estimated acceleration when motion capture is phase corrected to account for system latency. The validation results show a decaying sinusoidal curve that is characteristic of an underdamped system and IMU sensor readings agree closely with phase-corrected motion capture data. Note the scaling difference between absolute and error plots and absolute signal plots. The insert is for reader reference to the relevant local body axes. IMU sampling rate: 710 Hz. Motion capture sampling rate: 240 Hz.

**Figure 8 sensors-24-01805-f008:**
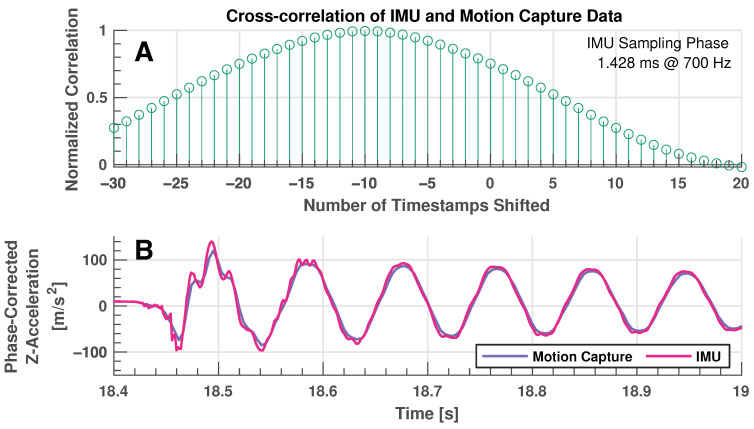
Results of cross correlation analysis. (**A**): Stem plot showing signal correlation as a function of samples lagged between IMU z-acceleration and motion capture estimated acceleration for the impulse validation test. (**B**): Comparison of IMU z-acceleration and motion capture estimated acceleration that has been phase-corrected for the impulse validation test. This plot is in direct comparison to Figure 7C that has not been phase corrected—note the dramatic improvement in correlation between the signals. Magenta line: IMU data, purple line: motion capture data.

There is a constant phase shift in motion capture data that is independent of the direction or frequency of movement (although a constant delay in signal would naturally affect higher frequencies to a greater extent). The source of this constant phase error is likely from the motion capture system. The particular setup utilized reports of a minimum latency of 4.2 ms in the product specifications, but additional latency is often introduced through sub-optimal setup and computing. The motion capture software (Motive, version 3.0.1) estimates a minimum system latency of approximately 5 ms in the position data that are recorded. Additional latency is further introduced in the way that IMU and motion capture data are time-synced. The higher-speed IMU system (700 Hz) indicates that data recording has commenced through an infrared LED. This indicator is likely to occur between two motion capture measurement frames recording at 240 Hz (4.17 ms period). This means that the triggering method induces a delay in the motion capture data a anywhere from 0 ms (trigger occurs as the current frame is captured) to a full frame (4.17 ms, trigger occurs immediately after the previous frame is captured). The estimated motion capture latency in combination with additional latency induced by the triggering method explains the majority of the latency observed in the motion capture data.

Furthermore, it would be concerning if the IMU measurements lagged the ground truth motion capture data, as this would indicate a potential delay in the data collection of the developed ski sensing tool. What we see in the data is a delay in motion capture measurements instead. This kind of delay in data is a common source of error when employing motion capture systems. In [39], the authors compared the latency of wireless motion sensors and a Qualisys optical motion capture system for triggers in offline performance and musical expression, where each system involves sampled data at 100 Hz. They reported an average latency of 23.5 ± 4.7 ms while measuring a single trigger event, accounting for the increased sampling rate of our motion capture system at 240 Hz (and assuming that latency scales linearly with frame rate) that would decrease their expected latency to 9.79 ± 2 ms with an optimal equipment setup, which is in line with our observed latency.

### 5.3. In-Field Experiments

Here, we describe the results of passing in-field IMU data (underneath the boot) through the KF algorithm described in Section 4. The algorithm performs remarkably well considering that a low-cost MEMS sensor is used. Figure 9A shows Euler angle estimates for an entire time series dataset of a representative run down a ski slope. Yaw angle estimates show limited drift over time, an impressive result, as this estimate tends to yield inaccurate results over larger timescales because of accumulated sensor bias during integration [40]. Furthermore, pitch and roll estimates agree with qualitative estimates from a camera attached to the skier’s boot, as can be seen in Figure 9B–E which show distinctly different ski poses and their corresponding KF estimates. This figure shows a representative dataset from a set of over 15 downhill runs collected over various days of testing that included different snow conditions, skiers and ski capabilities. Figure 9 shows still frames from a full video of a representative downhill run. The full video showing camera footage captured from the boot of the skier, along with dynamical measurements and KF estimates can be found in the Appendix A.

Figure 10 is used as confirmation that sensor data are realistic when compared to what is truly experienced. The figure shows the vertical z-acceleration measurement for the five distributed sensors on the forebody of the ski, with subfigure A showing data from an entire downhill run and subfigures B–D showing shorter 0.5 s windows of these data. Sensors located further from the ski binding have higher magnitudes, which is an intuitive result, as these locations typically experience higher flexural deformations and, as a result, higher accelerations. Furthermore, we also observe that in some cases sensor readings closer to the ski binding lag readings closer to the tip which may be an indication of a disturbance in the terrain (this lagging effect is not related to the lag in motion capture data discussed in Section 5.2). The tip sensor first encounters a disturbance followed by each successive sensor as the length of the ski passes over the disturbance. This lagging behavior is even more evident when the data from Figure 10 are compared with data from the laboratory impulse test in Figure 7B, where similar waveforms are observed, but the lagging characteristic seen during in-field testing is not present in the laboratory environment. This is because there is no disturbance in the laboratory environment that affects each sensor sequentially as there may be during in-field testing. The sensing system’s ability to extract spatio-temporal information directly related to the structural dynamics of the ski during in-field testing is one of the major advantages of distributed sensing and allows the extraction of far richer datasets than a single sensor could yield.

**Figure 9 sensors-24-01805-f009:**
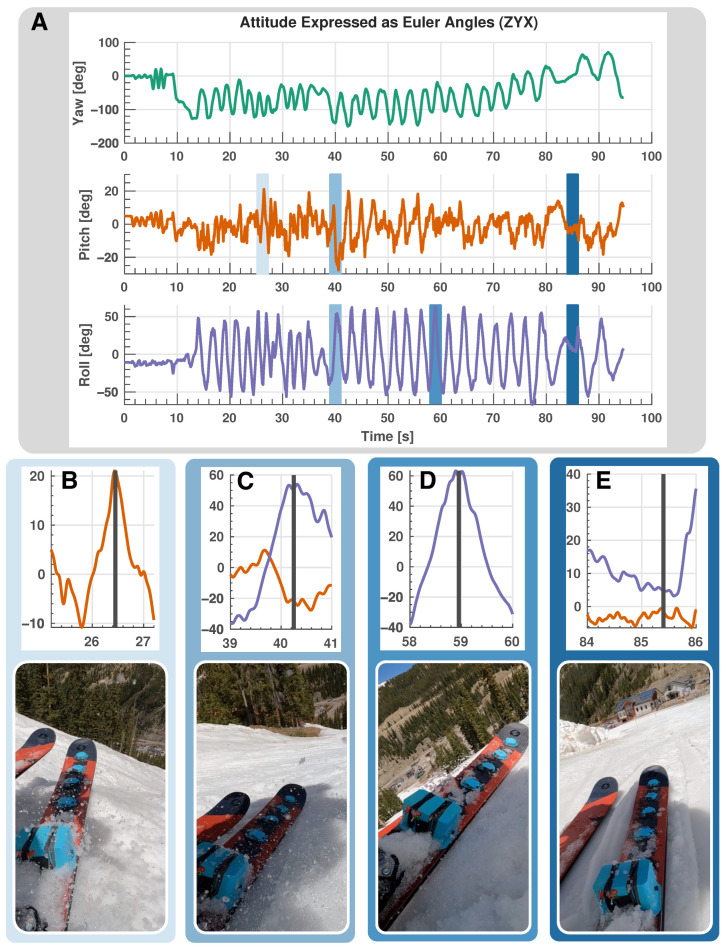
In-field ski testing results. Green lines: yaw angle, orange lines: pitch angle, purple lines: roll angle. (**A**): KF estimation of attitude expressed in Euler angles for the entire run lasting around 95 s. (**B**–**E**): Specific snapshots of the data in (**A**), together with camera footage of the system in action. The vertical black line indicates the exact instant when the video snapshot is extracted. (**B**): High-pitch-angle state. (**C**): High-roll-angle and low-pitch-angle state. (**D**): High-roll-angle state. (**E**): Leveled state with low roll and pitch angles.

**Figure 10 sensors-24-01805-f010:**
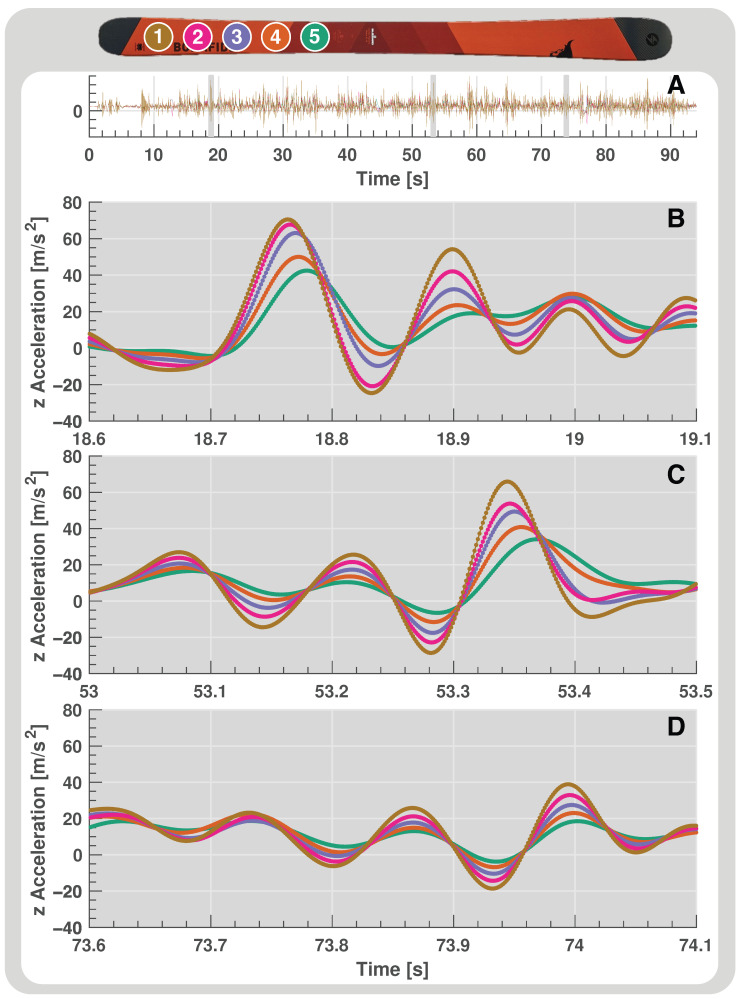
Vertical, z-acceleration readings from distributed sensors on forebody of an alpine ski during in-field testing. Brown line: IMU 1, magenta line: IMU 2, purple line: IMU 3, orange line: IMU 4, green line: IMU 5 (IMUs listed from tip to binding). (**A**): Time series of data for an entire downhill run. (**B**–**D**): The 0.5 s windows of the data shown in (**A**). The ski image indicates the sensor locations on the alpine ski. Sensors located closer to the tip have higher amplitudes due to higher levels of deflection when compared to sensors closer to the binding of the ski. Sensors closer to the binding appear to lag the sensors closer to the tip, which may indicate a disturbance in the terrain.

## 6. Discussion

The objective of this study is to create an accurate dynamic sensing system that can be used on alpine skis for in-field measurements. This sensing system needs to collect data from many points along the length of the ski at a high rate in order to detect multiple vibrational modes of the structure. We have shown that the developed system achieves all of these objectives while demonstrating the robustness qualities necessary for repetitive in-field testing in harsh conditions. Additionally, all technology is contained in a lightweight package located directly on the upper surface of the ski, allowing the user to freely use the ski in an unimpeded manner.

This system has a plethora of application areas due to its ability to collect rich and diverse datasets, both in a laboratory setting and during in-field testing. The first research domain that the authors are most excited to explore is that of ski equipment evaluation. The use of dynamic sensors as quantitative feedback information for ski designers is a relatively unexplored domain and data from the sensing system can be used to inform the equipment design loop for more rapid product development. Furthermore, the sensor suite may be useful to the equipment review community to complement qualitative review with quantitative data from in-field testing. Sensor data can also potentially be used in machine learning algorithms for ski/skier/snow classification and first-principle methods can be applied to investigate ski loading during in-field testing. Finally, there is potential to use the developed system for static laboratory testing, similar to [9], where static stiffness profiles are generated through the application of known loads on the ski. Sensor orientation information can then be used to calculate stiffness profiles along the length of the ski. Furthermore, the sensing of ski equipment can potentially be integrative with human motion sensing techniques, such as those referenced in [41,42], to create an overall performance evaluator for athletes where both athlete and equipment motion are evaluated.

Although the design of the distributed sensor suite achieved all of the design requirements, there are some limitations to the operation of the system and there are areas of improvement that can be explored. The most obvious improvement from a manufacturing point of view is that the custom IMU breakout boards should be outsourced for manufacturing. The tiny form factor of the IMU makes it prohibitively challenging to assemble the PCB board with conventional lab equipment and it would be more time- and cost-effective to outsource PCB manufacturing. Another area of potential improvement would be to explore additional signal termination strategies that are different to the serial resistors currently used. A major design hurdle for this system is signal integrity on the SPI bus at high transmission rates and different termination methods may yield more robust signal integrity for even higher data rates. Additional IMU sensors can also be added to the system to achieve a more continuous measurement of the ski dynamics, a somewhat trivial task with the modular system design. A simple improvement to this system would be more accurate synchronization of boot video footage with the sensor system. This is easily accomplished with an LED indicating the instant when data recording is initiated through the toggle switch (as is performed with the lab validation testing).

## 7. Conclusions

This study has detailed the implementation of a spatially distributed suite of robust IMU sensors on the upper surface of an alpine ski. This distributed sensing system is shown to yield accurate data when compared to a motion camera system that is considered to be the ground truth during laboratory validation testing. Not only did raw, dynamic sensor readings agree well with ground truth data, but KF-estimated orientation values also accurately aligned with motion capture information. This system has also undergone extensive testing (more than 15 isolated downhill runs) in a harsh outdoor alpine skiing environment where sensors are subject to cold and wet conditions and external mechanical shock. Sensors have proven to operate robustly in these in-field conditions, even when tested on different days by different skiers. Furthermore, the distributed sensors gather data at a high rate (>700 Hz), which yields rich, spatio-temporal datasets which can be used for more informed ski design and evaluation.

## Figures and Tables

**Figure 1 sensors-24-01805-f001:**
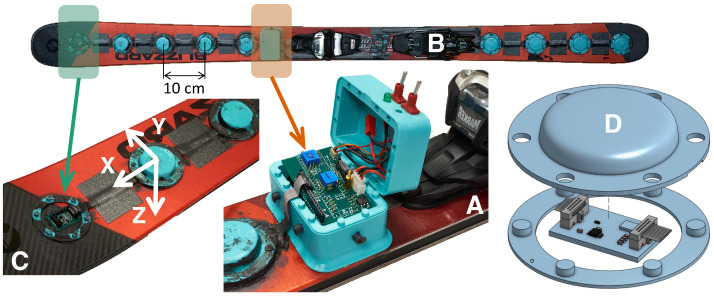
The distributed IMU sensors are attached to the upper surface of the ski. (**A**): The Central Processing Hub (CPH) that houses the battery, MCU, custom power and signal distribution board and user control toggle switches. (**B**): View of the sensing system from above the ski. Five sensors are placed on the forebody ahead of the CPH, one sensor underneath the boot and four sensors on the aftbody. Note the IMU breakout boards closest to the front tip and underneath the ski boot do not have a soft cover for demonstrational purposes. (**C**): An oblique view of the front tip of the ski with the local axis system used for data processing. (**D**): An exploded view of the sensor assembly model, including the breakout board, ribbon cable and soft protective covering.

**Figure 2 sensors-24-01805-f002:**
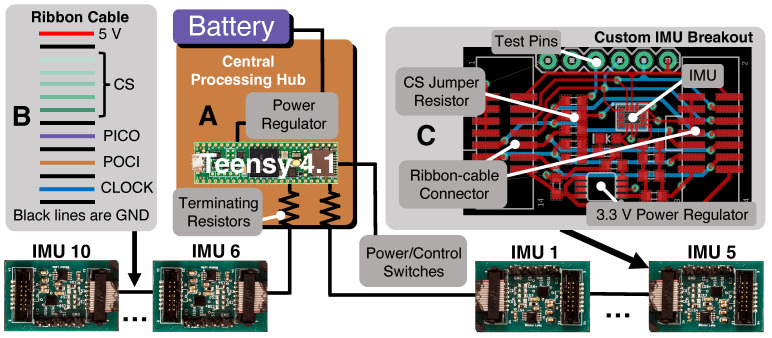
High-level overview of the electronic setup used for in-field data collection. (**A**): Overview of CPH functions: power regulation from the battery, communication with sensors and data collection and storage. (**B**): Ribbon cable signal layout, including 5V power, CS lines and SPI signal lines with ground lines for separation of signals to limit electromagnetic interference. (**C**): Layout of custom IMU breakout board, including ribbon cable connectors, independent dropdown power supply, CS jumper resistors and test pins.

**Figure 4 sensors-24-01805-f004:**
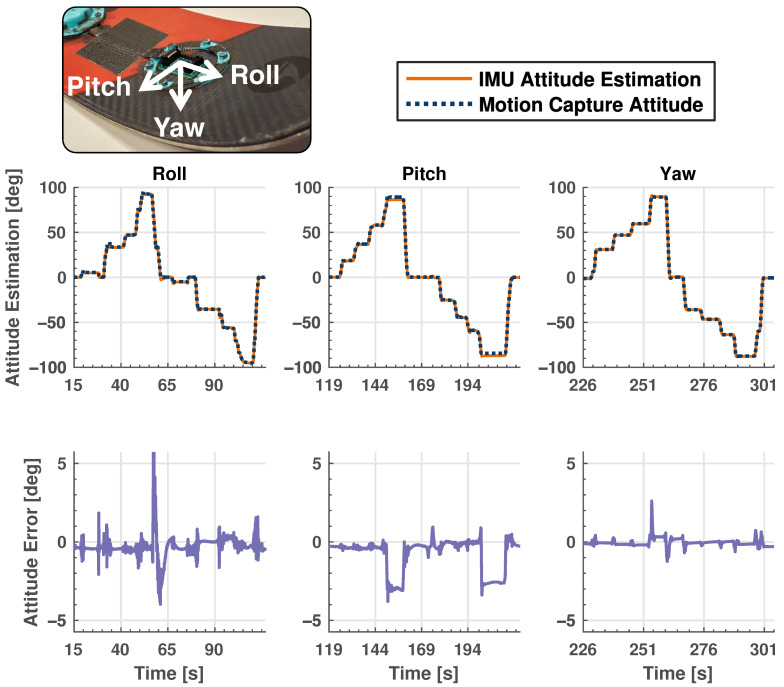
Results of quasi-static validation testing showing absolute orientation estimate data in the first row and error data in the second. Columns represent Euler angle representation about each of the three axes. Solid orange plots represent IMU data, while dashed, navy blue lines show motion capture data. This validation shows that the KF estimation tracks ground truth orientation for all three primary axes as they were subject to similar movements. Note the significant scaling difference between absolute and error plots. The insert is for reader reference to the relevant local body axes. IMU sampling rate: 710 Hz. Motion capture sampling rate: 240 Hz.

**Table 1 sensors-24-01805-t001:** Technical specifications of MEMS IMUs (TDK InvenSense MPU-42688-P) located on the upper surface of an alpine ski.

Term	Value
Gyroscope Noise	0.0028°/s$\sqrt{\mathrm{Hz}}$
Gyroscope Range	±2000°/s
Gyroscope and Accelerometer Resolution	16 bits
Accelerometer Noise	X, Y Axis: 65; Z Axis: 70 μg/$\sqrt{\mathrm{Hz}}$
Accelerometer Range	±16 g

**Table 2 sensors-24-01805-t002:** Definitions of terms used in the Indirect Complementary Kalman filter to estimate IMU attitude.

Term	Definition
${\hat{P}}_{e,k}^{-}$	A priori estimate covariance matrix
$Q_{k}$	Process noise covariance matrix (user-defined)
$y_{k}$	Innovation—determines the difference between expected and true error measurements
$z_{e,k}$	True error measurements
$S_{k}$	Innovation covariance matrix
$H_{k}$	Error measurement matrix that maps error measurements to error states
$K_{k}$	Kalman gain
$P_{e,k}^{+}$	A posteriori estimate covariance matrix

**Table 3 sensors-24-01805-t003:** Cross-correlation results of dynamic validation experiments. For all experiments, motion capture data lag IMU data by approximately 9 IMU measurement periods (12.86 ms).

Measurement	Timestamp Shifts for Maximum Correlation	R Value	Phase-Adjusted R Value	Percent Improvement
Gyroscope-X	−9	0.9941	0.9954	0.13
Gyroscope-Y	−9	0.9967	0.9978	0.11
Gyroscope-Z	−9	0.9932	0.9943	0.11
Accelerometer-X	−10	0.9974	0.9995	0.21
Accelerometer-Y	−9	0.9893	0.9910	0.17
Accelerometer-Z	−9	0.9911	0.9935	0.24
Impulse	−10	0.5316	0.9901	86.25

## Data Availability

Data from static and dynamic laboratory validation experiments and in-field testing can be found at the following repository: https://www.kaggle.com/datasets/leopoldbeuken/distributed-imu-sensors-on-an-alpine-ski, accessed on 8 January 2024.

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
