# Peer review of "Distributed IMU Sensors for In-Field Dynamic Measurements on an Alpine Ski"

_sensors, 2024, doi:10.3390/s24061805_

Round 1

Reviewer 1 Report

Comments and Suggestions for Authors

1. It is necessary to write the extended form of abbreviations in the first appearance (for example, MEMS)

2. Page 6 line 213: "who performed performed a variety" should be corrected

3. The two methods used in the laboratory evaluation to record data have different recording rates (frequency), how have you managed this issue?

4. Can the authors explain why the error in Figure 4 has increased at second 50 (approximately) but not at second 100 (approximately)? While probably similar movement conditions have happened.

5. Is the error obtained for the acceleration in Figure 6 an acceptable error to analyze or reconstruct the athlete's motion in a simulator? Can you provide a comparison table of the accuracy of other methods?

Comments on the Quality of English Language

There are some typos in the text that should be corrected

Reviewer 2 Report

Comments and Suggestions for Authors

The paper presents a technological application of Inertial measurement units. In my opinion in the motivation is not clear at least as regards the end user: how could an end user gather useful information for selecting the most suitable product from the proposed measurement system? How is it possible to correlate the measurement data with the ski behavior, considering also the influence of important factors such as snow conditions, skiing style and experience, slope grade, skier mass and muscle power?

Even if interesting as an application from the scientific point of view it presents few interest since no advances are actually proposed. Moreover the approach presents some criticalities that should be solved. For example it seems that 6 dof IMU are used in the experimentation, while currently available IMUs have 9 or more dof. The presence of a magnetometer is recommendable as authors declare as possible improvement.

The design process should be described starting from the specs required by the application considered: which are the maximum acceleration /angular rates foreseen during skiing? Which are the frequencies involved? What is the required sample rate? Is 700 Hz required in some way or is it just the maximum sampling frequency achievable by the system?

The laboratory validation is perhaps the more interesting part from the scientific point of view, but is not completely developed in the paper. The dynamic tests should be better organized considering the values required by the application, then they should be  characterized for example with the frequencies involved in the movements, the trial duration, the maximum amplitude for rotation rate and acceleration. As an example consider that the 5Hz cutoff filtering of the optoelectronic data (gold standard) limits the validation of the system. It is a low frequency, typical for gait analysis, that probably is really below the required bandwidth for a skiing test.

Reviewer 3 Report

Comments and Suggestions for Authors

1, What reason to make this research? Any novelty? It seems that most researchers are on the sensor componnet instead of the system.

2, As to IMU sensors, does the author consider the flexible sensor? How about the noise of the system?

3, The author consider the alpine ski application? However, it seems that the measurement seems tough, how overcome this issues about the real measrement

4, What reason to make kalman filter, how about the other filter? 

5, What is the differencec of the lab test and the filed test, what is the news between them?

6, Does the author consider the data processing , such as machine learning to the system on Figure 8?

7, How about the repeatance of the results to Figure 8?

8, please cite more papers to support the motion sensing background, such as Machine learning-assisted gesture recognition based on wearable self-powered tactile sensing, Sensors and Actuators: A. Physical. 365, 114877, 2024

9, What is the news of this research? Is the point on the application or test method?

Comments on the Quality of English Language

I am fine with the language

Round 2

Reviewer 2 Report

Comments and Suggestions for Authors

In this revision Authors have considered Reviewer comments and modified accordingly the paper. Now critical points have been corrected, details added and the paper quality has improved. I consider the paper acceptable in this new form.

Reviewer 3 Report

Comments and Suggestions for Authors

The authors did answer all my issues. I have no more questions